# Lactic Acid Bacteria as Mucosal Immunity Enhancers and Antivirals through Oral Delivery

**Assad Moon [1], Yuan Sun [1], Yanjin Wang [1], Jingshan Huang [1], Muhammad Umar Zafar Khan [2] and Hua-Ji Qiu [1,*]**

1 State Key Laboratory of Veterinary Biotechnology, Harbin Veterinary Research Institute, Chinese Academy of Agricultural Sciences, Harbin 150069, China
2 Institute of Microbiology, University of Agriculture Faisalabad, Punjab 38040, Pakistan
* Correspondence: qiuhuaji@caas.cn; Tel.: +86-451-5199-7170

**Abstract:** Mucosal vaccination offer an advantage over systemic inoculation from the immunological viewpoint. The development of an efficient vaccine is now a priority for emerging diseases such as COVID-19, that was declared a pandemic in 2020 and caused millions of deaths globally. Lactic acid bacteria (LAB) especially *Lactobacillus* are the vital microbiota of the gut, which is observed as having valuable effects on animals' and human health. LAB produce lactic acid as the major by-product of carbohydrate degradation and play a significant role in innate immunity enhancement. LAB have significant characteristics to mimic pathogen infections and intrinsically possess adjuvant properties to enhance mucosal immunity. Increasing demand and deliberations are being substantially focused on probiotic organisms that can enhance mucosal immunity against viral diseases. LAB can also strengthen their host's antiviral defense system by producing antiviral peptides, and releasing metabolites that prevent viral infections and adhesion to mucosal surfaces. From the perspectives of "one health" and the use of probiotics, conventional belief has opened up a new horizon on the use of LAB as antivirals. The major interest of this review is to depict the beneficial use of LAB as antivirals and mucosal immunity enhancers against viral diseases.

**Keywords:** lactic acid bacteria; exo-polysaccharides; mucosal immunity; antivirals; probiotics

## 1. Introduction

The animal and human body carry a variety of microorganisms, which are collectively referred as microbiota. The term 'gut microbiota' describes the collection of microorganisms colonizing the gastrointestinal tract (GIT). All of the microbes of the enteric microbiota often have a symbiotic correlation with their host, providing nutrients and protection from invading pathogenic organisms. However, opportunistic enteric pathogens can also be present in the enteric microbiota. These microorganisms cause infections when the host is immunocompromised [1]. Probiotics, which are living microorganisms, have a positive impact on the host by re-establishing the gut microbiota when taken orally in appropriate amounts; meanwhile, synergistic combinations of probiotics and prebiotics are called synbiotics [2]. The animal and human mucosal surfaces are exposed to various pathogens that cause diseases. So, prevention of the pathogen's entry onto the mucosal surfaces is critical for disease prevention. Probiotics such as lactic acid bacteria (LAB) prohibit the entry of viruses and other pathogens and significantly benefit animal and human health [3].

Depending on the characteristics of bacteria, LAB are Gram-positive, non-motile, non-spore-forming, generally rods or cocci, and have a strong tolerance to low pH, are facultative anaerobic except *Bifidobacteria*, which are obligatory anaerobes, and catalase-negative organisms that produce lactic acid by the degradative metabolism of carbohydrates [4]. LAB have been classified into more than 60 genera, most of which are used as probiotics, including *Lactobacillus*, *Lactococcus*, *Leuconostoc*, *Pediococcus*, *Streptococcus*, *Enterococcus*, and

*Weissella,* and phylogenetic classes such as *Bacillus*, *Clostridia*, *Enterococcus, Streptococcus*, and *Mollicutes* [2,5,6]. As probiotics are essential for a good and healthy life, LAB may have many beneficial effects by improving the intestinal microbiota balance [7–10].

Many LAB strains have been identified as possessing multifunctional characteristics such as high fermentation capability and can modulate the immune system against invading pathogens [11]. Most of the LAB species are considered probiotics; however, some of the LAB species, such as *Streptococcus mutans*, are serious pathogens of periodontal-associated diseases such as dental caries. It is also responsible for infective endocarditis (IE), which primarily occurs in cases with underlying heart disease [12,13]. The immunomodulatory effects and escalation of mucosal immunity by LAB may be accomplished by generating more mucin in the mucosa, developing a biofilm to mask the receptors for the attachment of viruses, and the activation of dendritic cells (DCs) [14]. Along with these events, the production of cytokines such as interleukin (IL)-6, IL-12, and gamma interferon (IFN-$\gamma$) and the activation of natural killer (NK) cells are responsible for the clearance of pathogens [14]. In the present review, we outlined and addressed the significance of LAB as mucosal immunity enhancer and antivirals, and presented new research and development objectives for probiotic-based oral vaccinations for emerging viral diseases.

## 2. LAB as Immunomodulators and Mucosal Immunity Enhancers

The immune system, which consists of the acquired and innate immune systems, works to neutralize invading viruses and other pathogens. Researchers reported that DCs play an important role in bridging innate and adaptive antiviral immunity. Numerous viruses are continually attacking the body. Epithelial surfaces, such as the skin and the mucosal linings of the digestive, respiratory, and urogenital tracts, which are home to DCs, are the first line of defense against pathogens, especially viruses [15]. When these barriers are breached, pathogens are captured by DCs, which are activated and attach to lymphoid organs where the proper specialized immune responses are initiated [15]. Mucosal immunity is the capacity to induce the protective immune response within mucosae where pathogens enter and initiate infections [16,17]. Animals and humans could initiate both systemic and mucosal immunity by recognizing pathogens as foreign objects for their neutralization. The difference between mucosal immunity and systemic immunity is the production of secretory immunoglobulin IgA (sIgA) which is more resistant to protease enzymes [18,19]. For protective mucosal immunity, participation of all kinds of mucosal immune cells are necessary for producing protective IgA antibodies. This process can be divided into entrance sites, where the pathogens adhere to the mucosal surface, and effector sites, where the plasma cells make antibodies that trigger a local immune response, as shown in Figure 1 [16,20].

The LAB strains significantly impact on the process of DCs' activation and the subsequent immunological responses. It has been demonstrated that murine DCs can respond differently depending on the strain of LAB, and this is exacerbated further by the fact that these responses can vary even amongst DC subtypes [21–23]. It has been reported that *Lactobacillus* modulates the maturation and function of DCs, macrophages, and CD$^{4+}$FoxP3$^+$ regulatory T cells (Tregs) as well as the differentiation of CD$^{4+}$CD$^{8+}$ and CD$^{4+}$FoxP3$^+$ T cells in Peyer's patches (PPs) [24,25]. The counterattack of pathogens is carried out by specialized DCs of mucosa in the mesenteric lymph nodes also called membrane-associated lymphoid tissues (MALTs). These lymphoid tissues are located beneath the mucosal epithelium of the intestine. These MALTs are similar to peripheral lymph nodes with an abundant supply of B cells and M cells for capturing the invading pathogens [26]. LAB could also initiate the cellular response by differentiation of DCs, and production of cytokines, that could favor the differentiation of näive T cells into Tregs, which are a specialized T cell subpopulation with specific regulatory mechanisms that inhibit the core components of adaptive and innate immune responses [27]. Tregs can drive the depression of an excessive response of effector T cells either by Th1, Th2, or Th17 and maintain mucosal immune homeostasis [10]. Differentiated DCs perform a significant role in the triggering of the immune

system against challenging viruses by attaching to them. These DCs are mainly located in the MALTs of the mucosal membrane of the intestine along with some draining lymphoid nodes in the mucosal membrane of the gastrointestinal tract. Plasmacytoid DCs (pDCs) and conventional DCs (cDCs) are the types of DCs presenting at the mucosal membrane. The pDCs are less commonly found in the blood circulation, the mucosal membrane of GIT, and the lymphoid tissue that produces IFN-$\alpha$ [28]. The DCs in the mucosal membrane are classified into CX3CR1$^+$CD103$^+$ DCs with fractalkine (FKN) receptors and CX3CR1$^+$ DCs. Among these DCs, CX3CR1$^+$ DCs have long stellate extensions which elongate from epithelial cells to the antigen found in the lumen of the gut and they usually do not migrate to another place. The mucosal immunity is thought to be organized within the MALTs, thus the antigen must be transported from the lumen to the MALTs by DCs for Tregs to initiate the immune response [29]. As a result of priming, a cascade of cytokines such as TGF-$\beta$, IL-6, IL-10, IL-12, IL-23, and other molecules are produced. These cytokines started a cascade of other interleukins' production and priming of T-helper cells to produce Th1, Th2, Th17, and other T regulatory cells for the neutralization of invading pathogens [30]. Gut microbiota dysbiosis increases the susceptibility of an individual to various diseases. Emerging evidence suggests that LAB are beneficial for the control of RV and SARS-CoV-2 infections. Probiotics are known for restoring stable gut microbiota through the interactions and coordination of the intestinal innate and adaptive immunity [31]. The researchers have reported on the effective protection of LAB against gastrointestinal viruses that originated from clinical cases in humans [32]. The activation of antiviral peptides and the production of mucin by intestinal epithelial cells, and the activation of the local innate immune system lead to an increase in sIgA antibodies for neutralization of the challenge [32]. RV infection deteriorates the mucosal barrier of the GIT [33]. In the clinical cases of RV infection, when *Lacticaseibacillus rhamnosus* GG (LGG) is administered orally, it could prohibit diarrhea caused by RV infection by mucosal immunity enhancement [33]. The treated cases of LGG reduced the adverse effects of RV on the barrier function in the GIT of piglets, improved relatively the intestinal microbiota, lowered autophagy, increased apoptosis of epithelial cells in the ileum, and retarded the viral multiplication in the intestinal epithelial cells (IECs) [33,34]. It has also been shown that combinations of probiotics and immunization work together to effectively change the gut microbiota. An oral RV vaccine's immunogenicity is increased by *Lactobacillus acidophilus*, which also improves the production of IgG and IgA antibodies. Probiotics such as *Lactobacillus rhamnosus* GG and *Bifidobacterium lactis* Bb12 also modulate dendritic cell responses via distinct Toll-like receptor (TLR) signaling, and function as immunostimulants for the RV vaccine [31].

LAB given orally travel down into the intestinal tract surviving through the stomach and are entrapped in the mucus layer secreted in the villi of the small intestine. Lactobacilli from enteric cells could come into contact with the mucosal epithelium. IgA that is secreted by sensed plasma cells in the epithelial membrane is secreted via the IgA receptor into the gut lumen and could be a superintended factor in bacterial presence. Pathogens that come in contact with the apical surface of the mucosal membrane might be sensed by DCs that can capture the viruses by entailment through their protrusions between enterocytes without breaching the integrity of the epithelial layer. The PPs found in the enteric wall are major contact sites where pathogens and antigens are prone to attach to enteric cells. M-cells in the epithelium transport pathogens present inside the lumen to the membrane-associated lymphoid tissue (MALTs) where pathogens are neutralized. DCs that are present in the area of the PPs can uptake and phagocytose viruses and transport them to the MALTs, where they can directly modulate immune responses that are activated by the potential pathogens.

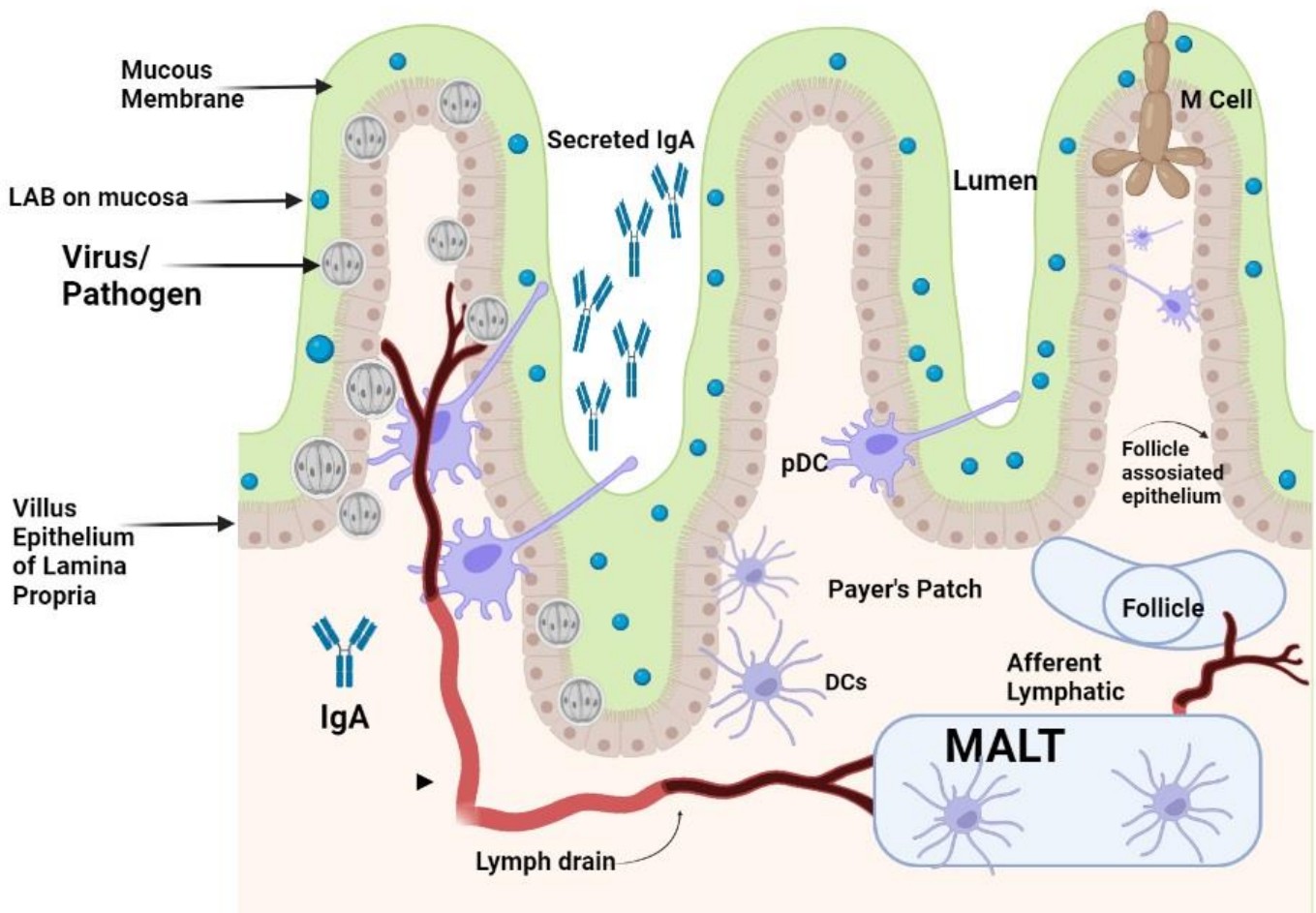

**Figure 1.** Effect of orally administered LAB on activation of gut-induced mucosal immunity.

### 2.1. Immunomodulatory Responses of LAB on Cytokines and Immune Cells

Many kinds of cytokines are produced by mucosal epithelial cells of the intestine such as TGF-$\beta$, IL-10, IL-12, IL-25, and IL-33, immune cells including NK cells, antigen-presenting cells (APCs), and T cells which are regulated by LAB resulting in enhanced mucosal immunity [32]. These cytokines and cells retard the viral invasion and increase innate immune response [32]. *L. plantarum* 06CC2 increased the immunomodulatory effect by escalating the mRNA expression of IFN-$\gamma$ and IL-12 in the PPs. In PPs, macrophages release IL-12 to activate NK cells after recognizing the viruses in the intestinal lumen. Activated NK cells modulate macrophages by releasing IFN-$\gamma$, which is an activator for macrophages and DCs. Both IFN-$\gamma$ and IL-12 have significant effects on antiviral mucosal immunity. Macrophages and NK cells mutually promote virus clearance in addition to direct phagocytosis of viruses and the virus-infected cells [35]. The molecular mechanisms of LAB as probiotics in animals for the production of mucosal immunity are still undefined. Researchers have conducted studies to explain the mucosal transcriptomic responses of healthy animals to the orally administered *Lactobacillus* strains. As expected, differential genomic expression was observed in the oral administration of *L. acidophilus* L10, *LGG*, and *L. casei* CRL-431 [36]. It is further reported that *L. acidophilus* L10 modulates IL-23 signaling and has a harmonious role in immune protection against RV infection. *L. acidophilus* L10 upregulates the proclamation of Th1-specific IFN-induced chemokines, such as CXCL11 and CXCL10, and IFN-stimulated genes (ISGs) [32].

## 2.2. PRRs-Associated Immunomodulation

Pattern-recognition receptors (PRRs) on outer mucosal surfaces have substantial interactions between pathogens and hosts. These are specialized attachment surfaces by which host cells recognize pathogens. The PRRs that are modulated by LAB are comprised of Toll-like receptors (TLRs). The members of TLRs can recognize invading pathogens such as viruses, and bacteria [37]. LAB can promote an innate immune response through the Gram-positive cell wall peptidoglycan and lipotechoic acid, which activate the TLR-2, nucleotide-binding oligomerization domain (NOD)-like receptors (NLRs), and C-type lectin receptors [38,39]. It has been reported that LGG is beneficial for diarrhea caused by RV infection through the TLR3 signaling pathway [37]. LAB-activated NOD2 and TLR2 receptors mediate the significant innate immune response to prevent RV invasions [37]. In addition, several studies have shown that the immunomodulatory activities of the probiotic mixture. For instance, LAB such as *Bifidobacterium infantis* R0033, *L. helveticus* R0052, and *Bifidobacterium bifidum* R0071 have significant effect on downregulating the proclamation of inflammatory cytokines, such as IL-6, IL-8, and IL-1$\beta$. The major impact on these cytokines includes upregulating the expression of TLR3 and mitogen-activated protein kinase and downregulating nuclear factor kappa B (NF-$\kappa$B) expression [40]. The EPSs produced by LAB also stimulate APCs, and activation of TLRs specifically through TLR2 and TLR4 signaling pathways [41–43]. The PRRs presenting on the outer cell surface could also serve as LAB receptors that can change cell signaling and transcription factors and are also responsible for inducing and enhancing cytokine production to counter invading pathogens.

Researchers reported that IECs sense the dsRNA of the virus via PRRs including RIG-I, TLR3, and MDA-5 [44]. After the identification of dsRNA by those receptors, innate cellular responses are initiated to neutralize infections. The initiating of PRRs in response to invading viruses gives rise to the production of chemokines, cytokines, IFNs, and ISGs that play significant roles in manifesting an antiviral environment and viral spread [44]. Many researchers have demonstrated that EPSs produced by LAB can favorably modulate PRRs-associated modulatory response in GIT by controlling the functions of IECs [45,46].

In GIT, IgA and sIgA play significant roles in the neutralization of invading viruses as the first line of defense. This IgA exhibits antiviral properties by collaborating with non-specific defense mechanisms [47,48]. In pig intestinal epithelium (PIE) cells, TLR3 activation by *L. rhamnosus* CRL1505 and *L. plantarum* CRL1506 significantly influence the production of IFN-$\gamma$, IFN-$\beta$, IL-8, MCP-1, and IL-6 [46,49]. It has been demonstrated that *L. rhamnosus* CRL1505 and *L. plantarum* CRL1506 also differentially regulate the expression of chemokines, cytokines, and adhesion molecules on the surface of PIE cells.

## 3. LAB as Antivirals

The animal and human mucosa are exposed to several viruses and pathogens. Mucosal surfaces are the path for the internalization of viruses and pathogens. Thus, preventing the virus's adsorption onto the mucosal epithelial surfaces is crucial for reducing disease development. LAB when administered orally neutralize the invading viruses and pathogens by preventing the attachment to mucosal surfaces and by production of several metabolites such as EPSs, bacteriocins, and ROS [10].

### 3.1. LAB-Produced EPSs as Antivirals

Like other bacteria, LAB also produce several types of EPSs made up of glucose and fructose as natural polymers of sugar and carbohydrates with their structural diversities and their applications are shown in Table 1 [50]. These EPSs are either bound loosely or released into the extracellular environment [51,52]. Many proteins and enzymes are involved in manufacturing EPSs that can be controlled by their active genes in the nucleus. Generally, the LAB-produced EPSs are beneficial for the host. Based on chemical characterization, EPSs are constituted by sugar components linked with each other by different ratios of monomers of glucose, galactose, rhamnose, mannose, and xylose to form homo-polysaccharides

(HoPSs) and hetero-polysaccharides (HePSs) [53]. EPSs produced by LAB possess a lot of potential as antivirals against emerging diseases such as COVID-19 and possess health benefits for animals and humans [54]. In animals and humans, EPSs have a significant impact, with anti-viral, immunomodulatory, mucosal immunity-enhancing, anti-cancer therapy, cholesterol-lowering, anti-biofilm, anti-hypertensive, anti-ulcer, anti-tumor, and anti-oxidant properties [54].

**Table 1.** Types of exo-polysaccharides, compositions, and applications.

| EPSs | Chemical Composition | Producer Strain | Application | References |
|---|---|---|---|---|
| | | **HoPSs** | | |
| Mutan | α-1,3 glycosidic linkage of the monomers of glucose. | *Streptococcus mutans, Leuconostoc,* and *Lactobacillus* species | Water-soluble HoPSs used as a starter culture. | [55] |
| Dextran | α-1,6 glycosidic linkages of the monomer of glucose. | *Leuconostoc, Lactobacillus,* and *Streptococcus* species | As an adjuvant, emulsifier, anti-coagulant, blood flow enhancer, and cholesterol-lowering agent. | [13] |
| Reuteran | α-1,4 and α-1-6 glycosidic linkage of the monomer of glucose. | *Lactobaciluus reuteri* | Water-soluble HoPSs used in bakery. | [13] |
| Alternan | α-1,6 and α-1,3 glycosidic linkage of the monomer of glucose. | *Lactobacillus mesenteriodes* | Highly water-soluble, low viscosity, and very less resistant to being hydrolyzed, used as probiotics. | [56] |
| Levan | β-2,6 and β-2,1 linkage of the monomer of fructose. | *Streptococcus salivarius, L. mesenteroides, L. sanfranciscensis,* and *S. mutans* | Cholesterol-lowering properties, natural adhesives, anti-tumorous effects, and are non-toxic to animals and humans. | [10,56] |
| Inulin | β-1,2 glycosidic linkage of the monomer of fructose. | *S. mutans* JC2, *Leuconostoc citreum, L. reuteri,* and *Lactobacillus johnsoni* | Indigestible, probiotics, prevent the attachment of viruses by reducing gastrointestinal wall pH, use as a vehicle for targeted drugs in colon carcinoma, and substitute for fat in food products. | [10,56] |
| | | **HePSs** | | |
| Keiferan | An equal proportion of glucose and galactose in linkage. | *Lactobaccilus kefiranum, Lactobacillus parakefir,* and *L. beijerinck, L. kefiri* | Wound healing, cholesterol-lowering medicines, anti-hypertensive drugs, tumor growth retardation medications, and enhancing mucosal immunity by IgA production. | [10,56] |
| Gellan | A tetra-saccharide structure is composed of 60% glucose, 20% rhamnose, and 20% glucuronic acid. | *Sphingomonas paucimobilis* ATCC 31461 | Use as a thickening agent or an adhesive in foods and as an agar substitute. | [57] |
| Xanthan | The backbone is made up of glucose and side chains consist of mannose-glucuronic acid-mannose. | *Xanthomonas campestris* | Moisture retention in wound dressings, suspension agents, and emulsion stabilizers in the food industry. | [58] |

The mode of action of the antiviral properties of LAB is accompanied by four main steps: viral absorption hindrance; internal uptake of viral particles into the lymphoid tissue; antiviral substance production; and immunomodulation in the mucosa [59]. The antiviral mechanism of LAB is multifactorial and irreversible [60]. Primarily, LAB may bind to the viruses and mask the binding sites of the virus surface and fusion proteins, thereby preventing viruses from entering the host cells [60]. Secondly, LAB may destroy the viral envelope and lyse the virions. Besides the action on human immunodeficiency virus (HIV), *L. crispatus* also have a remarkable inhibitory effect on HSV [61]. Researchers reported that when *L. delbrueckii* subsp. *bulgaricus* OLL1073R-1 in fermented yogurt was

consumed orally, the titer of influenza virus was decreased substantially, while the activity of NK cells and production of IgA and IgG were significantly increased [62]. LAB such as *L. delbrueckii* subsp. *delbrueckii* OLL1073R-1 in vitro showed a significant propensity to bring about an enhanced expression of IFN-$\gamma$, IFN-$\beta$, MxA, and RNase in intestinal epithelial cells (IECs) when stimulated with poly (I:C). MxA are the major inhibitory proteins in the antiviral activity that could safeguard against several viral infections [62,63]. Many researchers reported that LAB destroy viruses through an absorption or trapping mechanism [64,65]. For instance, vesicular stomatitis virus (VSV) can easily be entrapped by *Lactocaseibacillus paracasei* A14, *L. paracasei* F19, *L. paracasei/rhamnosus* Q8, *Lactiplantibacillus plantarum* M1.1, and *Lactiplantibacillus reuteri* DSM12246 [21]. According to a growing body of studies, probiotics that are administered orally help to strengthen the body's defenses against respiratory virus infections such as SARS-CoV-2 and influenza virus [31]. The gut microbiota is principally modified by LAB's well-established capabilities against SARS-CoV-2 by inhibiting the proliferation of opportunistic bacteria [31]. Scientists studied the antiviral properties of LAB against respiratory viruses such as influenza and SARS-CoV-2 as shown in Table 2. The transmission of SARS-CoV-2 from human to human mainly occurs via the respiratory route from an infected person [3,66]. During entry into the type 2 epithelial cells of the respiratory mucosa, SARS-CoV-2 uses the spike (S) glycoprotein to attach to the angiotensin-converting enzyme 2 (ACE-2) receptors found on the host's respiratory mucosal surfaces. The tendency of the SARS-CoV-2 is to attach to type 2 epithelial cells found on lungs' GIT and kidneys, which act as a reservoir for the virus. Lungs have a larger area covered with type 2 epithelial cells, which could be the explanation to the lungs being more prone to SARS-CoV-2 infection as compared with other body organs [3]. Oral administration of probiotics enhances the immunological response in the host through balancing the (Th1/Th2) immune responses and has a significant role in treating or alleviating pathologies associated with SARS-CoV-2 [3]. The immune response is carried out by differentiation of CD8$^+$ T-lymphocytes into cytotoxic T-lymphocytes, which are efficient in destroying the viral-infected cell. CD4$^+$ T-lymphocytes cells can also be differentiated into Th1, which starts phagocytosis by macrophages and NK cells and neutralizes the pathogens [3]. The strains belonging to *Lactobacillus* and *Bifidobacterium* genera possess immunomodulatory and antiviral activities. Research based on clinical trials suggested that probiotic strains, such as *L. casei, B. lactis* Bb-12, *L. rhamnosus* GG, *B. longum*, *L. plantarum*, and *L. casei* strain Shirota, remarkably decreased the upper respiratory tract infections, flu-like symptoms, and antibiotic-associated diarrhea by more than 70% [3,67,68].

Researchers reported that COVID-19 patients from Zhejiang province (China) were depicted with GIT infections such as vomiting, diarrhea, and nausea. The RNA of SARS-CoV-2 is detected in the lower abdomen such as the stomach, intestines, rectum, and feces of patients [3,69]. According to studies, SARS-CoV-2 promotes the manufacture of inflammatory cytokines, the growth of DCs, and the generation of type I IFNs, which inhibit virus growth and hasten virus phagocytosis by APCs [70]. Major histocompatibility complex (MHC-I) extends the antigenic peptides of coronavirus, which were previously recognized by virus-specific cytotoxic T cells (CTLs). MHC-I molecules are primarily responsible for the antigen presentation of SARS-CoV-2, but MHC-II is also involved [71]. Clinical cases with SARS-CoV-2 symptoms depicted microbiota dysbiosis with fewer probiotics such as *Lactobacillus* leading to the impaired immunological functions of the host [72,73]. The significant response exerted by probiotics in clinical cases of SARS-CoV-2 patients is to enforce and maintain the integrity of the junction between enterocytes, in this way attachment and entry of SARS-CoV-2 is impaired, as well as lowering the chances of being infected with SARS-CoV-2 [74]. It has been reported that *L. rhamnosus* CRL1505, *L. casei* DK128, *L. gasseri* SBT2055, and *B. subtilis* 3 were found to be the most immune-boosting probiotics to be explored further in the prevention and treatment of SARS-CoV-2 infection [3,75,76]. It has been reported that probiotics, such as *L. rhamnosus GG* can help to improve GIT microbiota and promote homeostasis, by enhancing and improving the antiviral defense by the production of a cascade of pro-inflammatory cytokines (IFN-$\gamma$, IFN-

$β$, IL-1, IL-6, IL-12, IL-18, IL-33, TNF-$γ$, and TGF-$β$) and chemokines (CCL2, CCL3, CXCL8, CXCL9, and CXCL10) in the form of a "cytokines storm" in systemic and respiratory infections of COVID-19 patients, as shown in Figure 2 [71,72,76].

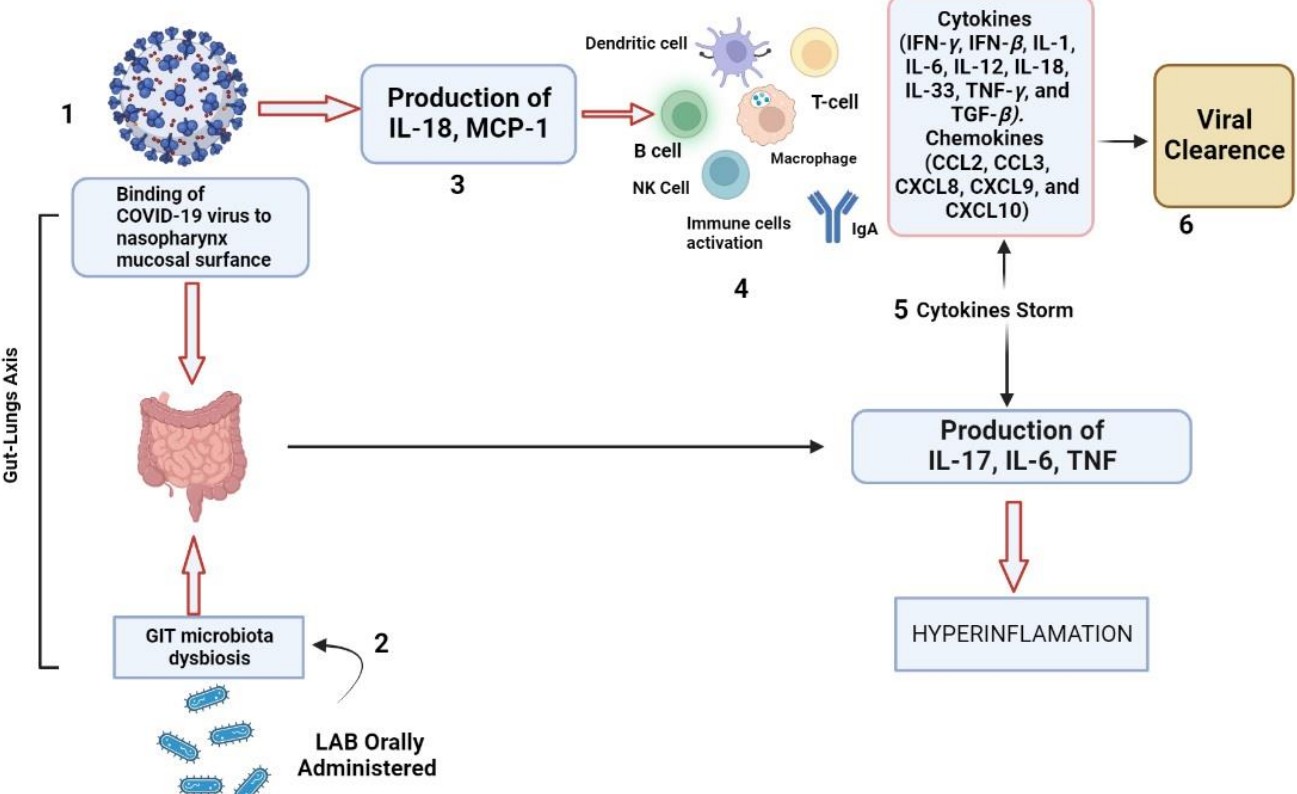

**Figure 2.** Antiviral effects of LAB against SARS-CoV-2. Schematic representation of gut–lungs axis with SARS-CoV-2 viral infection on the respiratory surface (1) causing GIT microbiota dysbiosis. (2) When LAB are administered orally, they lead to the production of IL-18 and MCP-1 (3) which lead to the maturation of immune cells (4) such as regulatory T cells, B-cells, NK-cells, and macrophages and the production of IgA and various cytokines (5) such as IL-12, IL-21, IFN-$γ$, and IFN-$β$, which enhance viral clearance (6).

The LAB, when used at the initial occurrence of the disease, have been proven to provide effective protection against viral diarrhea caused by gastrointestinal viruses by shortening the duration of diarrhea [2]. For the antiviral properties of LAB, the polymer group of sulphated polysaccharides plays a significant role against pathogens [77]. The macromolecules of sulphated polysaccharides were found to have strong antiviral activities [78]. However, the chemical structure and polymerization of polysaccharides have not yet been carefully studied [77]. The researchers found that *L. plantarum* (R315) and *Bifidobacterium bifidum* (WBIN03) have beneficial effects on gut microbiota by their secreted EPSs as compared with other LAB species such as *Escherichia coli*, *Shigella sonnei*, *Cronobacter sakazakii*, *Staphyloccocus aureus*, *Listeria monocytogenes*, and *Salmonella enterica* subsp. *enterica* serovar *typhimurium* [77].

**Table 2.** Antiviral activities of lactic acid bacteria against various infectious viruses.

| Strain/Vehicle | Targeted Disease | Route | Immune Response | References |
|---|---|---|---|---|
| *Enterococcus faecium* L3 | Influenza virus (H1N1, H3N2) | In vitro MDCK (Madin-Darby canine kidney cell line), and in vivo (female mice). | Enterocin B stimulates IFN production and boosts the innate immune response. | [79,80] |
| *L. reuteri* ATCC 55730 | *Enteroviruses* and *Coxsackie viruses* CA6 and CA16 | Intestinal Caco-2 cells and in vitro (skeletal muscle RD-cell line culture). | Immune modulation of chemokines and inflammatory cytokines production. | [81] |
| *L. casei* DK128 | Influenza virus (H3N2 and H1N1) | In vivo administration in female mice. | Rapid induction of IgG2a and IgG1 antibodies and induction of innate immune response and production of cytokines. | [80,82] |
| *L. acidophilus*, *L. rhamnosus*(LGG) | Enteroviruses | In vitro. | Induction of IL-12 and IgA, modulation of Th1 immune response. | [80,83] |
| *S thermophiles*, *L. plantarum* | Influenza virus | In vitro. | Boost up of Th1 immune response and induction of IL-12 and IgA. | [21] |
| *L. lactis* | HIV-1 | In vivo by the administration in mouse through intra-gastric route with Cholera toxin as an adjuvant. | Enhanced fecal Antibodies and serum antibodies. | [84,85] |
| *L. lactis* | Rotavirus | Administered in mice by the intragastric route. | Production of antibodies against rotavirus infection. | [85,86] |
| *L. casei* | SARS-CoV | Administered in mice by intragastric and intranasal routes. | Modulation of mucosal IgA antibodies and serum antibodies. | [85,87] |
| *L. lactis* JCM5805 | IFV | Human | Increased expression of IFN-$\alpha$ and ISGs. | [88] |
| *L. gasseri* SBT2055 | IFV | Mouse | Increased expression of ISGs. | [89] |
| *L. acidophilus* | HIV-1 | Murine BALB/C | Increased Intestinal and Vaginal Epitope-Specific IgA B cells. | [90,91] |
| *L. acidophilus* ATCC 4356 | H9N2 | DCs | Stimulation of type-I IFNs signaling pathway. | [40] |
| *L. casei* | Porcine epidemic diarrhea virus (Core neutralizing epitope) | Murine BALB/C | Increased Intestinal, Vaginal, Nasal, Ocular, and Serum IgA levels. | [92] |
| *L. casei* | Porcine rotavirus (VP4 capsid protein) | Murine BALB/c | Increased Intestinal, Vaginal, Nasal, Ocular, and Serum IgA. | [91,93] |
| *L. plantarum* | Avian influenza (hemagglutinin antigen) | Murine BALB/C | Increased CD4+ T Cell IFN- (MLN), IL-4 (MLN, Splenic), IL-17 (MLN, Splenic) and CD8+ T Cell IFN-$\gamma$ (MLN, Splenic). | [91,94] |
| *L. lactis* *L. plantarum* | Human papilloma virus (E7) | Murine C57BL/6 | Increased serum IgG; Increased GAL, IgA Increased IFN-$\gamma$ | [91] |
| *L. plantarum* | NDV (Hemagglutinin-neuraminidase) | Chicken | Increased splenic and peripheral blood CD4+ T cells. | [91,95] |

### 3.2. LAB-Produced Peptides as Antivirals

LAB-produced anti-microbial peptides (AMPs) have increased researchers' interest due to their therapeutic potential and broad-spectrum antimicrobial properties against viruses, bacteria, fungi, and protozoa [96]. Bacteriocins are one of the bacterial-derived

AMP [96–98]. Bacteriocins are versatile in the mode of action as antivirals [96]. The LAB-produced bacteriocins could lead to the blocking of the viral particles by competitively attaching to host receptors, accumulating viral particles, and inhibiting the viral replication cycle. The bacteriocins could form the aggregation of viral particles and inhibit the multiplication process of viruses [99]. The bacteriocins secreted by *Lactobacillus beijerinck* showed virucidal activities on the influenza virus [100]. Enterocin B produced by *Enterococcus* was shown to retard the cytopathic effects of H1N1 and H3N2 influenza virus in Madin-Darby canine kidney (MDCK) cells [79].

### 3.3. Reactive Species Produced by LAB as Antivirals

Reactive species (RS) produced by LAB include reactive oxygen species (ROS) and reactive nitrogen species (RNS), which lead to a highly oxidative environment and inhibit the viral replication process [101]. Hydrogen peroxide ($H_2O_2$) produced by the *Lactobacillus* species also plays a vital role in antiviral and antibacterial activities [101]. This chemical metabolite protects the host as an antiviral agent against HIV-1 and HSV-2. $H_2O_2$ extracted from the vaginal strain of *Levilactobacillus brevis* that retard the replication process of the HSV-2 [101]. It is also observed that these metabolites suppress T-lymphocytes' activation, resulting in lymphocyte sustainability in viral infections [101]. Previous studies on the antiviral effects of *Lactobacillus*-based probiotics have focused on the production of $H_2O_2$, but advanced studies have shown that EPSs produced by LAB are also antiviral factors produced by *Lactobacilli* in the vaginal mucosa [102,103]. In addition, lactic acid is the final product of carbohydrate metabolism and an important metabolite of *Lactobacillus* used for the neutralization of HIV [104–106].

## 4. LAB as Mucosal Vaccine Vectors

To maintain the good health of animals and humans, LAB have been introduced as a vector in animal feed and human food as beneficial organisms to prevent many lethal viral and bacterial diseases by modulating the immune system, as shown in Figure 3 [5]. Diseases caused by various gastrointestinal viruses remain a big challenge for farmed animals and for humans [107]. Vaccination is the most important option to prevent viral infections in farm animals, but differences between pandemics and vaccine strains make vaccination less effective. Moreover, vaccine development for novel viral strains is a difficult task. *Lactobacilli* have been used as vehicles for the delivery of vaccines to counter many viral diseases [108]. The choice of LAB as a vaccine vector is based on a variety of characteristics that render them very appealing as a possible means of vaccine delivery. Dietary LAB organisms have a very long history of safe administration through the oral route [109]. Additionally, LAB are able to colonize cavities such as the mouth, the urogenital, and GIT, where they play a critical role in maintaining a balanced normal microflora. In addition, LAB have an absence of lipopolysaccharides (LPS) in their cell wall that virtually eliminates the risk of endotoxic shock and survival inside the stomach due to acid resistance [109]. The commensal and dietary types of *Lactobacillus* strains are used as inherent vaccine vectors that give beneficial effects to animals and humans [110].

Many researchers have reported that *Lactobacillus gasseri* is the most advantageous species, and is considered the model organism to be used as a vaccine vector, because of the unchallenging manipulation of its genome. This has made *L. gasseri* more beneficial for biotechnological use, covering a range from the production of recombinant proteins to the expression and delivery of modified chimera and bioactive molecules to the mucosal surfaces [108,111]. Characteristics of LAB such as high resistance to the acidic environment of the stomach, the ability to remain in the GIT without colonizing, less immunogenicity, and the lack of lipopolysaccharides in its cell wall, which reduces the chances of endotoxin shock, making such organisms highly versatile to be used as vectors, including in immunization programs [109,112].

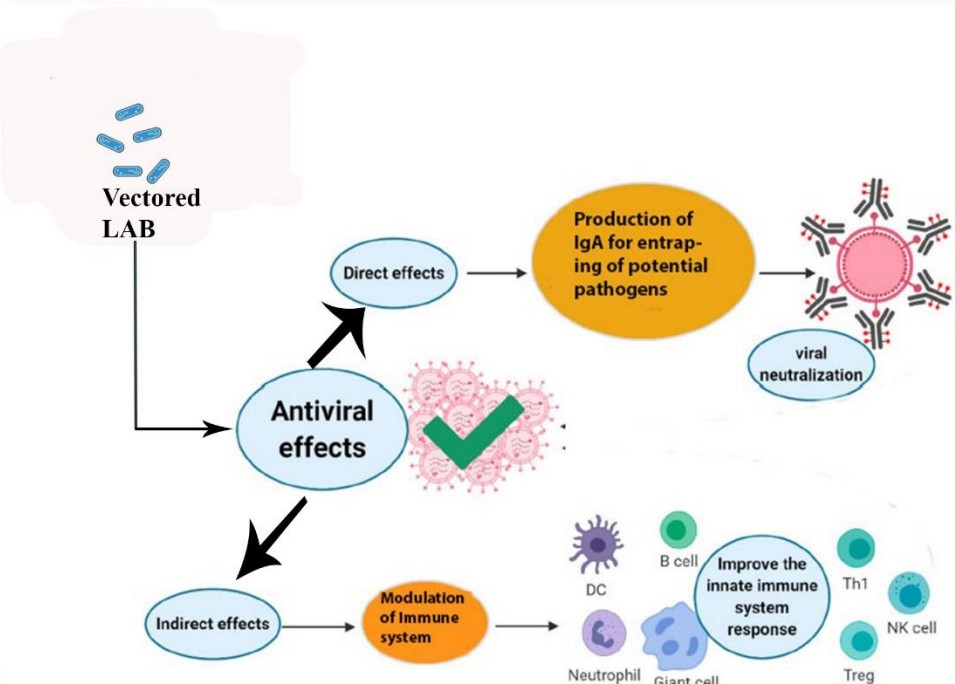

**Figure 3.** Direct and indirect effects of vectored lactic acid bacteria delivered orally in the enhancement of mucosal immunity. Lactic acid bacteria (LAB) when used as probiotics orally display direct and indirect effects on the health of animals and humans by preventing the attachment of potential pathogens to the mucosal surface of the intestine. LAB also modulate the innate immune system to produce antibodies and immune cells for the neutralization of viruses.

In pigs, the composition of the gut microbiota might change the host's immune response against invading viruses and other pathogens. Similar patterns have been seen in various viral infections, including the African swine fever virus and enteric viruses [113,114]. The production of various kinds of anti-viral peptides by LAB has also been reported by many researchers against viral infections [113,115]. The mechanism of antigen delivery to targeted DCs has tremendous potential for new-age vaccine development. LAB such as *L. lactis, L. acidophilus, L. gasseri*, and *L. casei* have great potential as a vector for the delivery of molecules orally to induce a mucosal immune response and production of IgA for many viral and pathogenic diseases. This advancement has a great leap to the conventional process of attenuating pathogens for vaccine development.

LAB have conserved pathogen-associated molecular patterns (PAMPs) such as peptidoglycans, cell wall polysaccharides, lipoteichoic acid (LTA), surface-associated adhesion molecules of Gram-positive bacteria, and lipoproteins which are anchored in the cell cytoplasm membrane [19,116]. It should be taken into account that various strains of LAB differ in their immune regulatory properties, which can have significant roles in intrinsic use as vectors. In particular, their ability to attach to the mucosal surfaces is a principal characteristic.

*4.1. Suitability of LAB as Vectors*

LAB have become increasingly significant in therapeutic uses such as anti-viral, immunomodulatory, anti-inflammatory, anti-oxidant, anti-tumor, anti-diabetic, enhanced colonization of pathogens, anti-hypertensive, and cholesterol-lowering actions, as shown in Figure 4.

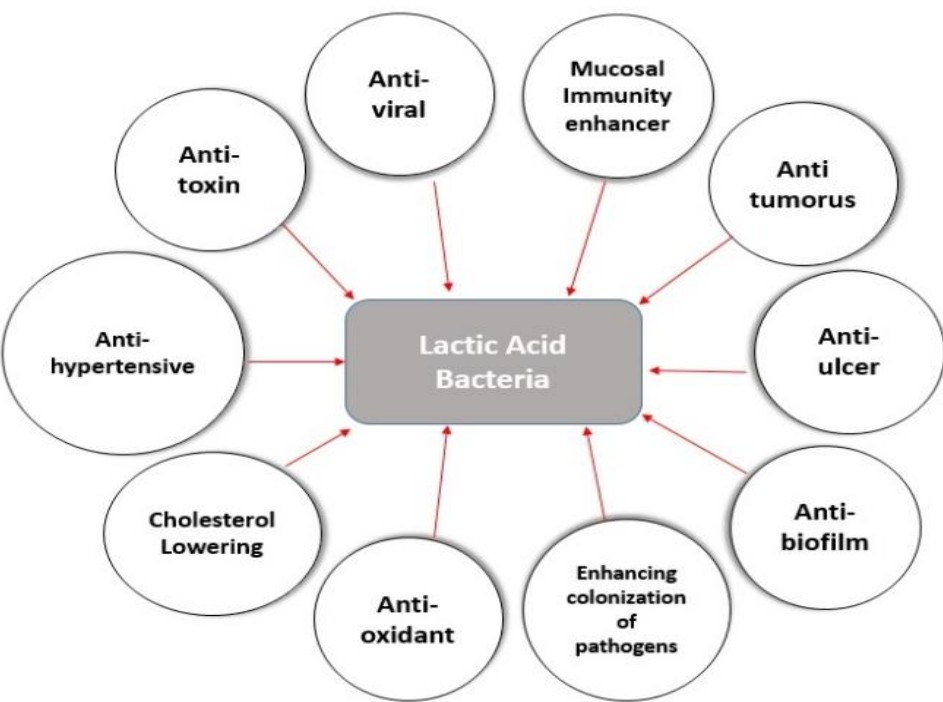

**Figure 4.** Health and nutritional benefits of lactic acid bacteria in the schematic diagram. LAB have a number of beneficial health effects on animals and humans. LAB typically colonize the GIT and then reinforces the host defense systems with its anti-microbial, immunomodulatory, anti-inflammatory, anti-oxidant, anti-tumor, anti-viral, anti-diabetic, anti-ulcer, and cholesterol-lowering properties; probiotic microorganisms have now become incredibly valuable in therapeutic applications against emerging viruses.

Commensal LAB strains can give healthful effects with the capability of sticking to the wall of the GIT, urethra, and vagina. Dietary strains lead to struggling with the microbiota in GIT, which can result in antigen disclosure for a longer duration. Both in vitro and in vivo techniques have been applied to screen sticky *Lactobacillus* strains to be used as vaccine vectors. Data have indicated that the colonization of LAB in GIT is site-specific [110]. The optimal expression of gene sequences, targeting sequences, transcription, and translation in LAB are strain-dependent. Recent advancements in genomic sequence included the development of recombinant plasmid expression vectors with enhanced stoutness and integrated genomic systems with targeted specific loci. These developments are mostly contingent on various non-replicative or chimeric conjugative transposons [110]. This approach may lead to components that pose an antibiotic resistance marker, with inactivated chromosomal genes necessary for immune regulation [110]. Autophagy is one of the significant areas of research by oncologists nowadays. This phenomenon is based on the type of cancer cells involved. Some cells of the body have been subjected to autophagy as a result of treatments and some other tumorous cells may be treated with some other different forms of cell death. It has been demonstrated that the LAB-produced EPSs can regulate the autophagy gene *Beclin-1* and also interrelate with apoptosis-related genes [117].

*4.2. Limitations and Risks of LAB as Oral Vaccines*

LAB are generally recognized as safe (GRAS) and also contain various essential properties that make the way more difficult for their effective use as advantageous vaccine carriers, but a lot of risks persist that need to be disclosed and resolved before their use is in the best interests of humans.

The vaccine which consists of a live attenuated organism contains antigen encoding genes that are either present on the plasmid or amalgamated on an organism's DNA. In both circumstances, the concerns with the protection, such as the eventual result of the

genetically modified plasmid in the vaccine, must be appraised. However, the vaccine use of a specified host range plasmid replication system retards the horizontal pass-on of the plasmid to other microbes and prevents unwanted perseverance of the plasmid. Research should evaluate which cells absorb and express the DNA, the destiny of the DNA in that cell, and the time required in which the DNA persists in the cells. The dose of the modified DNA should also be probed to clarify the defensible amount of plasmid gathered peripherally to the target cells. Another significant difficulty for the use of the LAB as a vaccine is the propensity of these bacteria to survive if released in a natural environment which is a highly arguable issue and safeguards must be rigorously maintained to avoid their spread. To reduce this risk, the use of an auxotrophic chimera should be a priority for LAB used as vectors. Without appropriate growing conditions, such an LAB chimaera cannot spontaneously proliferate in the environment.

## 5. Conclusions and Future Perspectives

Viral diseases are linked with dysbiosis of the intestinal microbiota leading to severe GIT infections. So, oral probiotic-based treatments are becoming significant in the prevention of viral infections. Probiotics can regulate host immunity and counteract the "cytokine storm" production during viral infections such as COVID-19. However, probiotic-based treatment against novel viruses such as SARS-CoV-2 infections in the field is still an open research question. Viral infections in the respiratory tract are one of the rapidly surging global diseases with high morbidity rates. The intensity can range from a mild upper respiratory tract infection to a severe chronic infection of the mucosal layer of the lower respiratory tract and multi-organ failure in some cases. Activation of the immune system is one of the best prophylactic techniques to lower the severity of such viral diseases. Oral administration of probiotics has various advantages such as strengthening the gut barrier function, balancing the composition of the gut microbiota, and initiation of protective immune responses against invading viruses and pathogens. Bacterial vector vaccines have been studied experimentally for more than a decade; however, interestingly, no live, recombinant bacterial vectors are available for commercial human or veterinary use. Various constraints exist, such as the safety of vaccine strains remains a major issue at the level of vaccinated individuals and environmental spread. To date, the results obtained with LAB are very encouraging as they show that these non-pathogenic, non-invasive bacterial vectors are capable of taking antigens to the mucosal and systemic immune systems initiating specific antibody responses in serum and secretions. While both GIT colonizers and non-colonizers seem to work equally well by the systemic and respiratory routes, the importance of colonization or adhesion in oral administration against viruses is still under appraisal. In the process of recombinant LAB vaccine development, various key points need to be addressed. It is important to pursue a detailed analysis of the immune response generated in relation to the mode of antigen presentation and the delivery route and to further improve the effectiveness of LAB as antigen carriers in order to compare them to the other bacterial vectors under development. It is also necessary to attain knowledge about the antigen production level in vivo, the adhesion mechanism of LAB with GIT walls, and the fate of LAB when administered orally. Last but not least, as in the case of probiotics, well-controlled studies have to be performed in humans or animals in order to clarify the colonization capacity of properly selected *Lactobacillus* strains and their interaction with the immune system and the endogenous microbiota of the host.

**Author Contributions:** Conceptualization and approval: H.-J.Q.; original draft preparation: A.M.; review and editing: Y.S., Y.W., J.H. and M.U.Z.K. All the authors equally contributed to this work. All authors have read and agreed to the published version of the manuscript.

**Funding:** This study was supported by the National Key R&D Program of China (Grant no. 2021YFD1801403).

**Institutional Review Board Statement:** Not Applicable.

**Informed Consent Statement:** Not Applicable.

**Data Availability Statement:** Not Applicable.

**Conflicts of Interest:** The authors declare no conflict of interest.

**Abbreviations**

LAB: Lactic acid bacteria; EPSs: Exo-polysaccharides; HoPSs: Homo-polysaccharides; HePSs: Hetro-polysaccharides; DCs: Dendritic cells; APCs: Antigen-presenting cells; GIT: Gastrointestinal tract; IECs: Intestinal epithelial cells; HIV: Human immunodeficiency virus; HSV: Herpes simplex virus.

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
