# Peer review of "Lactic Acid Bacteria as Mucosal Immunity Enhancers and Antivirals through Oral Delivery"

_2673-8007, doi:10.3390/applmicrobiol2040064_

Round 1

Reviewer 1 Report (Previous Reviewer 2)

  1. The manuscript " Mucosal Immunity Enhancement and Antiviral Potential of Lactic Acid Bacteria through Oral Delivery" has an interesting outcome. The authors have improved the manuscript. However, the manuscript needs to be improved for publication in Applied Microbiology. Need to revise before publication by English editing service.

    1. L51: Use reference: https://doi.org/10.1016/j.tifs.2020.03.019
    2. L76: Use reference: https://doi.org/10.1016/j.foodcont.2021.108276
    3. L83: Cite most recent article: https://doi.org/10.1039/D1FO03864H 

Author Response

Responses to the Reviewers’ comments on Applied Microbiology (ISSN 2673-8007)

Dear Professor,

Thank you very much for reviewing our manuscript. We greatly appreciate the valuable comments and suggestions. We have studied the comments carefully and revised our manuscript accordingly. Frankly, the reviews are very professional and constructive. A marked-up manuscript with changes highlighted in blue has been provided for your reference. A point-by-point response is appended as follows.

Reviewer # 1

  1. 1. L51: Use reference: https://doi.org/10.1016/j.tifs.2020.03.019.

Author’s Response: The reference has been added as per suggested in Line 53.

  1. L76: Use reference: https://doi.org/10.1016/j.foodcont.2021.108276.

Author’s Response: The reference has been added as per suggested in Line 77.

  1. L83: Cite the most recent article: https://doi.org/10.1039/D1FO03864H.

Author’s Response: The recent reference has been added as per suggested in Line 85.

  1. The authors have improved the manuscript. However, the manuscript needs to be improved for publication in Applied Microbiology. Need to revise before publication by English editing service.

Author’s Response: The manuscript was sent to a native English speaker. We have incorporated the suggestions in the manuscript.

Reviewer 2 Report (New Reviewer)

The article entitled as "Mucosal Immunity Enhancement and Antiviral Potential of 2 Lactic Acid Bacteria through Oral Delivery" is an interesting review article and well structured. However, I have following suggestions to increase the impact of the present review article. 

1. Line no. 11: Change to: The development of an effective vaccine is now a desire for emerging viruses such as SARS-CoV-2. 

2. Line no. 12: COVID-19 has become a pandemic in 2020.

Elaborate it more so it will not look odd in the abstract. 

3. Line no. 39 Probiotics such as LAB, provide the abbreviation of LAB at first time and then use the LAB in the whole manuscript. 

4. Line no. 58: Dendritic cells (DCs)

5. Line n0. 71: line of defense against these viruses.

I wonder which viruses?, mention some viruses here with appropriate references. 

6. Line no. 84, Only write DCs now.

7. Line no.111: T regulatory cells (Tregs) 

8. Line no. 93 to 96:

Reframe the information to highlight the functionality of the Tregs for more clarity as following:

that could favour the differentiation of naive T-cells into Tregs. Tregs are a specialized T cell subpopulation with specific regulatory mechanisms that inhibit the core components of adaptive and innate immune responses (Goswami et al., 2022). Tregs can drive the depression of an excessive response of effector T cells either by Th1, Th2, or Th17 and maintain mucosal immune homeostasis [10].

Goswami TK, Singh M, Dhawan M, Mitra S, Emran TB, Rabaan AA, Mutair AA, Alawi ZA, Alhumaid S, Dhama K. Regulatory T cells (Tregs) and their therapeutic potential against autoimmune disorders - Advances and challenges. Hum Vaccin Immunother. 2022 Dec 31;18(1):2035117. doi: 10.1080/21645515.2022.2035117. Epub 2022 Mar 3. PMID: 35240914; PMCID: PMC9009914.

9. Line no. 141: L. plantarum 06CC2 increased, 

Did not understand the above naming, check it please.

10. Line no. 188: L. rhamnosus CRL1505 and L. plantarum CRL1506

change the scientific names in italics

and check for this throughout the manuscript. 

11. Line no. 232-233: Reframe the information for the clarity.

12. Line 241-243: The tendency of SARS-CoV-2 virus is to attach with type-2 epithelial cells found on kidneys, GIT and lungs which act as a reservoir for the virus.

 I can not see the relevance of this information, please try to connect this with the previous sentences.

13. Line no. 264 increase the clarity and readability by fragmenting the information. 

14. Line no. 269: regulatory T cells (Tregs).

15. Line no. 400, The figure legend can be elaborated and descriptive.

16. Line no. 427: change to “cytokine storm” during viral infections such as SARS-CoV-2.  

17. Additionally, under the section 3.1 LAB-produced EPSs as antivirals

18. I believe a figure representing the possible antiviral effects of LAB against SARS-CoV-2 and their functionalities must be developed to increase the reach of the manuscript.

These recent articles can be used to incorporate in the review 

1. Montazeri-Najafabady N, Kazemi K, Gholami A. Recent advances in antiviral effects of probiotics: potential mechanism study in prevention and treatment of SARS-CoV-2. Biologia (Bratisl). 2022 Jun 28:1-18. doi: 10.1007/s11756-022-01147-y. Epub ahead of print. PMID: 35789756; PMCID: PMC9244507.

2. Nguyen QV, Chong LC, Hor YY, Lew LC, Rather IA, Choi SB. Role of Probiotics in the Management of COVID-19: A Computational Perspective. Nutrients. 2022 Jan 10;14(2):274. doi: 10.3390/nu14020274. PMID: 35057455; PMCID: PMC8781206.

Best Wishes

Author Response

Responses to the Reviewers’ comments on Applied Microbiology (ISSN 2673-8007)

Dear Professor,

Thank you very much for reviewing our manuscript. We greatly appreciate the valuable comments and suggestions. We have studied the comments carefully and revised our manuscript accordingly. Frankly, the reviews are very professional and constructive. A marked-up manuscript with changes highlighted in blue has been provided for your reference. A point-by-point response is appended as follows.

Reviewer # 2

  1. Line no. 11: Change to: The development of an effective vaccine is now a desire for emerging viruses such as SARS-CoV-2.

Author’s Response: The statement has been rephrased as suggested in (Lines 11-12).

  1. Line no. 12: COVID-19 has become a pandemic in 2020. Elaborate it more so it will not look odd in the abstract.

Author’s Response: The statement has been rephrased as suggested in (Lines 12-13).

  1. Line no. 39 Probiotics such as LAB, provide the abbreviation of the LAB the first time and then use the LAB in the whole manuscript.

Author’s Response: The abbreviation has been added as suggested in (Lines 38-39) and rechecked in the revised manuscript.

  1. Line no. 58: Dendritic cells (DCs).

Author’s Response: The abbreviation has been added as suggested in (Line 59).

  1. Line no. 71: line of defense against these viruses. I wonder which viruses? Mention some viruses here with appropriate references.

Author’s Response: The statement has been modified with an appropriate reference added in (Lines 73-75).

  1. Line no. 84, only write DCs now.

Author’s Response: The abbreviation has been added as suggested in (Line 86).

  1. Line no.111: T regulatory cells (Tregs).

Author’s Response: The abbreviation has been added as suggested in (Line 91).

  1. Line no. 93 to 96: Reframe the information to highlight the functionality of the Tregs for more clarity as follows: that could favor the differentiation of naive T-cells into Tregs. Tregs are a specialized T-cell subpopulation with specific regulatory mechanisms that inhibit the core components of adaptive and innate immune responses (Goswami et al., 2022). Tregs can drive the depression of an excessive response of effector T cells either by Th1, Th2, or Th17 and maintain mucosal immune homeostasis [10]

Author’s Response: Suggested information has been incorporated in (Lines 99-101).

  1. Line no. 141:  plantarum 06CC2 increased, did not understand the above naming, check it, please.

Author’s Response: We have rechecked the name, the given name is according to the scientific literature in (Line 159).

  1. Line no. 188: rhamnosus CRL1505 and L. plantarum CRL1506, change the scientific names in italics and check for this throughout the manuscript.

Author’s Response: Italicized the bacteria names in (Line 207) as suggested and checked throughout the manuscript.

  1. Line no. 232-233: Reframe the information for clarity.

Author’s Response: We have rephrased the sentence in (Lines 251-252).

  1. Line 241-243: The tendency of the SARS-CoV-2 virus is to attach with type-2 epithelial cells found on kidneys, GIT, and lungs which act as a reservoir for the virus. I cannot see the relevance of this information, please try to connect this with the previous sentences.

Author’s Response: We have connected the sentence in (Lines 251-255).

  1. In line no. 264, increase the clarity and readability by fragmenting the information. 

Author’s Response: We have rephrased the sentence in (Lines 258-263).

  1. Line no. 269: regulatory T cells (Tregs).

Author’s Response: We have changed the T-cells to Tregs in (Line 298).

  1. Line no. 400, the figure legend can be elaborated and descriptive.

Author’s Response: We have added the description in Figure 4 in (Lines 444-447).

  1. Line no. 427: change to “cytokine storm” during viral infections such as SARS-CoV-2.

Author’s Response: We have changed the sentence according to suggestions in (Line 473).

  1. Additionally, under section 3.1 LAB-produced EPSs as antivirals.

Author’s Response: We have added information in (Lines 229-230).

  1. I believe a figure representing the possible antiviral effects of LAB against SARS-CoV-2 and their functionalities must be developed to increase the reach of the manuscript.

Author’s Response: we have incorporated Figure 3 in the manuscript elaborates on the possible antiviral effects of LAB against SARS-CoV-2.

  1. These recent articles can be used to incorporate in the review,
    1. Montazeri-Najafabady N, Kazemi K, Gholami A. Recent advances in antiviral effects of probiotics: potential mechanism study in prevention and treatment of SARS-CoV-2. Biologia (Bratisl). 2022 Jun 28:1-18. doi: 10.1007/s11756-022-01147-y. Epub ahead of print. PMID: 35789756; PMCID: PMC9244507.

Author’s Response: We have added the suggested literature in the manuscript in (Lines 283-289).

  1. Nguyen QV, Chong LC, Hor YY, Lew LC, Rather IA, Choi SB. Role of Probiotics in the Management of COVID-19: A Computational Perspective. Nutrients. 2022 Jan 10;14(2):274. doi: 10.3390/nu14020274. PMID: 35057455; PMCID: PMC8781206.

Author’s Response: We have added the suggested literature in the manuscript in (Lines 120-122).

We have also checked the full text of the manuscript.

Round 2

Reviewer 2 Report (New Reviewer)

The authors have revised the manuscript sufficiently and incorporated the changes as per the reviewer's comments. I highly appreciate the changes and corrections made by the authors to improve their article. Just change the numbering of the figures, as there are four figures now. 

Best Wishes 

This manuscript is a resubmission of an earlier submission. The following is a list of the peer review reports and author responses from that submission.

Round 1

Reviewer 1 Report

1.     In table 2, page 8.  Edit admiration to be administration

2.     Figure 1, page 12. The picture looks too complex than it is supposed to, and is not well described. The picture should be modified and completely labeled.

3.     Page 14, line 3. Add comma after Bifidobacterim infantis R0033

4.     Page 16, line 6. The text of B. anthracis should be all italic.

5.     Page 17, lines 2-5. “Lactobacillus gasseri is the most advantageous species, and is considered the model organism used as a vaccine vector because of the unchallenging manipulation of its genome.  This has made L. lactis more beneficial”. Should it be referred to “L. gasseri.”, and not “ L. lactis”

6.     Page 18, double check figure number.

7.     Figure 2 on page 18, use arrow from Lactobacilli into antiviral effects and then separate arrow from antiviral effects into direct and indirect effects, respectively. 

8.     Figure 3, page 19. Change blue color to eye comfortable colors or use white color with black border.

9.     Heading of table 2 should be in sentence case.

Author Response

  1. Responses to the Reviewers’ comments on Applied Microbiology (ISSN 2673-8007)

    Dear Professor,

    Thank you very much for reviewing our manuscript. We greatly appreciate the valuable comments and suggestions. We have studied the comments carefully and revised our manuscript accordingly. Frankly, the reviews are very professional and constructive. A marked-up manuscript with changes highlighted in red has been provided for your reference. A point-by-point response is appended as follows.

    1. In table 2, page 8.  Edit admiration to be administration.

    Author’s Response: Thanks for suggesting this correction. We have changed admiration to administration in Table 2.

    1. Figure 1, page 12. The picture looks too complex than it is supposed to, and is not well described. The picture should be modified and completely labeled.

    Author’s Response: Proper labeling and modifications have been made in Figure 1.

    1. Page 14, line 3. Add a comma after Bifidobacterim infantisR0033.

    Author’s Response: We put a comma after Bifidobacterim infantis R0033.

    1. Page 16, line 6. The text of B. anthracisshould be all italic.

    Author’s Response: Corrected as suggested.

    1. Page 17, lines 2-5. “Lactobacillus gasseriis the most advantageous species and is considered the model organism used as a vaccine vector because of the unchallenging manipulation of its genome.  This has made L. lactismore beneficial”. Should it be referred to “L. gasseri.”, and not “L. lactis”

    Author’s Response: We have corrected L. lactis to L. gasseri.

    1. Page 18, double check figure number.

    Author’s Response: We have changed Figure 2 on page 18.

    1. Figure 2 on page 18, use arrow from Lactobacilli into antiviral effects and then separate arrow from antiviral effects into direct and indirect effects, respectively.

    Author’s Response: We have edited the Figure according to suggestions.

    1. Figure 3, page 19. Change blue color to eye comfortable colors or use white color with black border.

    Author’s Response: We have changed the color format of Figure 3.

    1. Heading of Table 2 should be in sentence case.

    Author’s Response: We have changed Table 2 heading to sentence case.

Reviewer 2 Report

The manuscript "Lactic Acid Bacteria as Mucosal Immunity Enhancers and Anti-virals through Oral Vaccination" has an interesting outcome. However, several reviews have very recently addressed the same topic and it is hard to see the novelty of this particular review and its benefit to the Applied Microbiology readership. Therefore, for publication in Applied Microbiology, the manuscript needs to be improved. Many grammatically problematic sentences were found throughout the manuscript, which must be checked and corrected precisely.  

  1. Can microbiota consist only of useful microorganisms in humans and animals? Verify your hypothesis with proper references.
  2. “Probiotics as living microorganisms when administered orally in sufficient quantity gave a beneficial effect on the host by re-establishing the gut microbiota, whereas synergistic combinations of probiotics and prebiotics are called synbiotics”-complex sentence and meaning are not clear. Suggested to rephrase the complete sentence.
  3. “primary entry sites in the body”- what do you mean?
  4. Is all LAB considered probiotics? Verify your hypothesis with proper references.
  5. Bifidobacteria, which are obligatory anaerobes, and catalase-negative organisms that produce lactic acid by degradative metabolism of carbohydrates”- put proper reference.
  6. Table 1: Use the full form of the bacteria name first time throughout the manuscript
  7. “Up to now, no definitive mechanism to explain the antibacterial action of the EPSs produced by the LAB has yet been identified against Gram-positive and Gram-negative bacteria”- this statement is not valid. Check this article https://doi.org/10.1016/j.tifs.2020.03.019
  8. Mention the full form of all abbreviated words (e.g., HIV, HSV) in the abbreviation section.
  9. “SARS-CoV-2 mainly occurs via respiratory route”-via should not be italic. Make corrections throughout the manuscript.
  10. There are two “Figure 1” captions. Authors are suggested to revise the manuscript.
  11. Authors are suggested to discuss the future perspective of probiotic treatment against viral infections.
  12. Many grammatically problematic sentences are found throughout the manuscript, which must be checked and corrected precisely. Authors are suggested to check the manuscript with professional English editing services.

Author Response

Responses to the Reviewers’ comments on Applied Microbiology (ISSN 2673-8007)

Dear Professor,

Thank you very much for reviewing our manuscript. We greatly appreciate the valuable comments and suggestions. We have studied the comments carefully and revised our manuscript accordingly. Frankly, the reviews are very professional and constructive. A marked-up manuscript with changes highlighted in red has been provided for your reference. A point-by-point response is appended as follows.

  1. Can microbiota consist only of useful microorganisms in humans and animals? Verify your hypothesis with proper references.

Author’s Response: The term 'gut microbiota' describes the collection of microorganisms that colonize the gastrointestinal tract (GIT). All the microbes of the enteric microbiota often have a symbiotic correlation with their host, providing nutrients and protection from invading pathogenic organisms. However, opportunistic enteric pathogens can also be present in the enteric microbiota. These microorganisms cause infections when the host is immunocompromised. These gut-lodging, opportunistic pathogens include Escherichia coli, Clostridioides difficile, and Enterococcus faecium. Moreover, species in the genus Bacteroides, particularly Bacteroides fragilis, are among the most notable anaerobic causes of GIT ailments. GIT infections are the most common infections caused by pathogenic enteric microbiota. Pathogenic enteric microorganisms such as cholera (Vibrio cholerae), salmonellosis (Salmonella sp.), dysentery (Shigella sp.), and ailments caused by Campylobacter jejuni, Yersinia sp., and Escherichia coli O157:H7 are the common pathogenic microorganisms of the gut microbiome.

  1. “Probiotics as living microorganisms when administered orally in sufficient quantity gave a beneficial effect on the host by re-establishing the gut microbiota, whereas synergistic combinations of probiotics and prebiotics are called synbiotics”-complex sentence and meaning are not clear. Suggested rephrasing the complete sentence.

Author’s Response: We have rephrased the sentence within the text “Probiotics, which are living microorganisms, had a positive impact on the host by re-establishing the gut microbiota when taken orally in appropriate amounts, whereas synergistic combinations of probiotics and prebiotics are called synbiotics”.

  1. “Primary entry sites in the body”- what do you mean?

Author’s Response: Primary entry sites mean most of the pathogens have an affinity to attach to mucosal surfaces to get entry inside the body to develop infections.

  1. Is all LAB considered probiotics? Verify your hypothesis with proper references.

Author’s Response: Most of the LAB species are considered probiotics but some of the LAB species such as Streptococcus mutans, is a serious pathogen of periodontal-associated diseases such as dental caries. It is also responsible for infective endocarditis (IE), which majorly occurs in cases with underlying heart disease.

  1. Bifidobacteria, which are obligatory anaerobes, and catalase-negative organisms that produce lactic acid by degradative metabolism of carbohydrates”- put proper reference.

Author’s Response: We have added the proper reference as suggested.

  1. Table 1: Use the full form of the bacteria name first time throughout the manuscript.

Author’s Response: Thanks for suggesting the corrections. We have changed the names of bacteria in Table 1 and checked the whole nomenclature throughout the manuscript.

  1. “Up to now, no definitive mechanism to explain the antibacterial action of the EPSs produced by the LAB has yet been identified against Gram-positive and Gram-negative bacteria”- this statement is not valid. Check this article https://doi.org/10.1016/j.tifs.2020.03.019.

Author’s Response: LAB-produced bacteriocins can also prohibit the bacterial biofilm formation of mostly Gram-negative bacteria by interfering with different enzymes such as RNA polymerase, aspartyl-tRNA synthetase, and DNA gyrase, thus damaging the bacterial DNA, RNA, and protein metabolism. For instance, microcin J25 can retard RNA polymerase [37], microcin C7/C51 can retard asp-tRNA synthetase [38], and microcin B17 can prohibit DNA gyrase [39].

  1. Mention the full form of all abbreviated words (e.g., HIV, HSV) in the abbreviation section.

Author’s Response: We have added the full form of abbreviated words in the abbreviation section.

  1. “SARS-CoV-2 mainly occurs viarespiratory rout via should not be italic. Make corrections throughout the manuscript.

Author’s Response: We have corrected ‘via’ throughout the manuscript.

  1. There are two “Figure 1” captions. Authors are suggested to revise the manuscript.

Author’s Response: We have revised Figure No. on the manuscript.

  1. Authors are suggested to discuss the future perspective of probiotic treatment against viral infections.

Author’s Response: We have added the section of the conclusion and future prospective of probiotic treatment against viral infections.

  1. Many grammatically problematic sentences are found throughout the manuscript, which must be checked and corrected precisely. Authors are suggested to check the manuscript with professional English editing services.

Author’s Response: The manuscript was sent to a native English speaker. We have incorporated the suggestions in the manuscript.

We have also checked the full text of the manuscript.

Round 2

Reviewer 2 Report

Thank you for your revision. I am satisfied with the revised version of the manuscript, which can be accepted.